# RadioAR: Autoregressive Modeling for Accurate Radio Map Estimation

## Abstract

Radio map estimation (RME) is a key enabler for environment-aware wireless systems, supporting tasks such as coverage planning, interference management, and context-aware resource allocation. Accurate RME is challenging because radio maps exhibit both smooth, large-scale variations (e.g., path loss and shadowing) and rapidly changing, high-frequency details induced by multipath propagation. This paper presents **RadioAR**, a multi-scale autoregressive framework that predicts radio maps from coarse to fine resolutions. We design a radio-map tokenizer based on *residual Laplacian pyramid decomposition* and *continuous tokens*, which preserves subtle signal variations while avoiding the quantization artifacts introduced by discrete tokenizers. On top of the tokenizer, RadioAR employs a conditional transformer to progressively refine token maps under building and transmitter conditions. Experiments on RadioMapSeer (IRT4) show that RadioAR achieves better accuracy than representative convolutional, transformer, GAN, diffusion, and Mamba baselines, while maintaining inference latency compatible with real-time deployment.

## CCS Concepts

• **Computing methodologies** → **Computer vision**.

## Keywords

radio map estimation, autoregressive modeling, multi-scale tokenization, transformer, wireless propagation, 6G

## 1 Introduction

Radio maps (Fig. 1(a)) describe the spatial distribution of radio-frequency metrics such as received signal strength (RSS), interference, and channel gain over a geographic area. They support a wide range of downstream tasks, including network planning [9], spectrum allocation [48], interference management [7], and path planning for autonomous platforms [43]. As wireless systems move toward higher carrier frequencies and denser deployments [32], accurate and low-latency radio map estimation becomes increasingly important.

Accurate radio map estimation (RME) is difficult in realistic environments because propagation is shaped by both multipath and blockage. Reflections, diffractions, and refractions create multiple propagation paths (Fig. 1(b)), leading to rapid spatial fluctuations in received power (small-scale fading; Fig. 1(c)). At the same time, obstacles attenuate the dominant paths and introduce slower but pronounced large-scale variations (shadowing; Fig. 1(d)). An effective RME model must therefore capture global trends and fine-grained textures within a tight latency budget.

*Conference'17, Washington, DC, USA*
2026. ACM ISBN 978-x-xxxx-xxxx-x/YYYY/MM
https://doi.org/10.1145/nnnnnnn.nnnnnnn

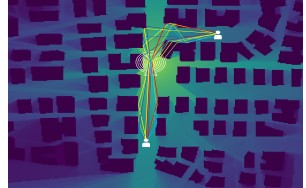

(a) Illustration of a radio map     (b) Multi-path propagation

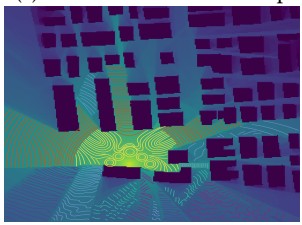 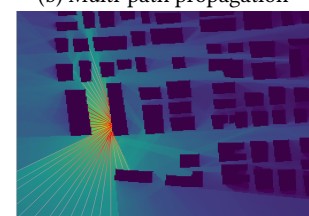

(c) Small-scale fading     (d) Large-scale fading or shadowing

**Figure 1: Spatial distribution of radio signals and challenges in radio map estimation. This figure presents an overview of radio signal behavior and the primary obstacles in radio map estimation arising from the intricate nature of electromagnetic wave propagation.**

Classical RME approaches often rely on physics-based propagation models [26] or empirical formulas [21]. While these methods provide interpretability, they are either too computationally demanding for large-scale deployment or insufficiently accurate when detailed scene information is unavailable.

Spatial statistical methods [1, 2, 4, 11, 13, 18, 24, 25, 27, 29–31, 33, 40] infer the full map from sparse measurements via structured interpolation or completion. These techniques are attractive when measurements are available, but their accuracy can degrade under sparse sampling and when the map exhibits strong multi-scale non-stationarity.

With modern compute, deep models have become the dominant approach for RME, learning mappings from scene context to radio maps. CNN-based estimators provide fast inference but can struggle to capture long-range dependencies due to limited effective receptive fields [17]. Transformer-based models improve global context modeling, yet often incur substantial compute and memory overhead at high resolution. Diffusion models can generate high-fidelity maps, but their iterative denoising procedure leads to high inference cost [10], which is undesirable for latency-sensitive settings. GAN-based approaches may offer efficient sampling but can be sensitive to training instability and mode collapse.

These trade-offs motivate a model class that is both *multi-scale* and *computationally efficient*. Visual Autoregressive Models (VAR) [34] predict token maps scale-by-scale from coarse to fine, and generate tokens in parallel within each scale, providing a favorable accuracy–latency profile. However, existing conditional VAR variants typically rely on discrete tokenizers, which introduce quantization

errors [20]. For RME, such discretization is particularly problematic as it acts as a low-pass filter, blurring the sharp spatial gradients caused by multipath-induced small-scale fading. Preserving these high-frequency transitions is a physical necessity for accurate signal mapping; otherwise, the resulting "blocking" artifacts can significantly bias downstream tasks.

We address this gap by proposing **RadioAR**, a multi-scale autoregressive framework tailored to radio map estimation. RadioAR combines a continuous-token, multi-scale tokenizer with a conditional transformer that progressively refines the map from coarse to fine resolutions.

Our contributions are:

(1) We introduce RadioAR, a multi-scale autoregressive model for radio map estimation that adapts VAR-style next-scale prediction to continuous-valued radio maps under building and transmitter conditions.

(2) We design a radio-map tokenizer based on residual Laplacian pyramid decomposition with continuous tokens, which preserves fine-grained variations while avoiding quantization artifacts.

(3) We evaluate RadioAR on RadioMapSeer (IRT4) against representative convolutional, transformer, GAN, diffusion, and Mamba baselines, and provide ablations that isolate the effects of model scaling and token representations.

## 2 Related Work

### 2.1 Radio Map Estimation

*Traditional Methods.* Traditional radio map estimation methods are primarily model-based, relying on physical models of radio wave propagation for prediction. Specific techniques include solving Maxwell's equations for simple geometries [12], utilizing empirical models such as ITU-R recommendations [28], and employing computational methods like finite-element analysis [8] and ray tracing [6, 47].

*Spatial Statistical Methods.* Spatial statistical methods estimate the complete radio map from measurements collected at spatially dispersed locations. Representative techniques in this category include Kriging interpolation [2, 25], radial basis function interpolation [11, 13], tensor completion [18, 27], and matrix completion [31, 40].

*Deep Learning Approaches.* Deep learning approaches are increasingly adopted for radio map estimation, offering a practical accuracy–latency trade-off. CNN-based models, such as RadioUNet [14] and FadeNet [23], leverage U-Net-style architectures for fast prediction at meter-level resolution. Graph neural networks (e.g., GNN-MDAR [41]) model non-Euclidean spatial relationships, while transformer-based models such as Radionet [36] use attention to capture global context. Generative formulations have also been explored: GAN-based methods (e.g., RME-GAN [49], ACT-GAN [3]) learn a conditional generator, and diffusion-based models such as RadioDiff [39] cast RME as conditional denoising to improve perceptual quality.

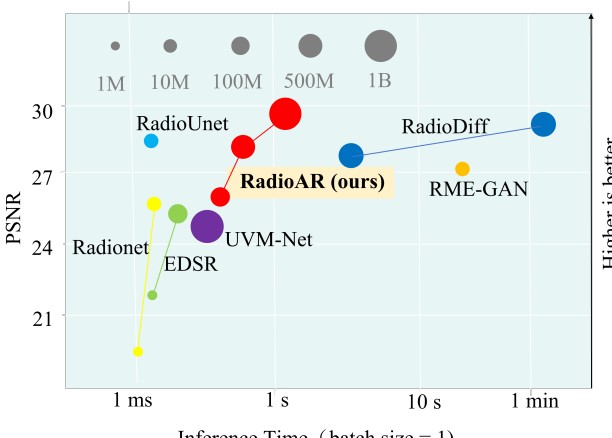

**Figure 2: Scaling behavior and inference speed performance of different model families on the IRT4 dataset benchmark.**

### 2.2 Autoregressive Models in Image Generation

Autoregressive (AR) models [44] have advanced image generation by treating images as sequences of discrete tokens or pixels, employing various architectures, often transformers, to predict subsequent elements. Early models like Vector Quantized Variational Autoencoder (VQVAE) [38] encode image patches into discrete tokens, modeled autoregressively using PixelCNN [37] in a raster-scan order. Vector Quantized Generative Adversarial Network (VQGAN) [5] enhances this by combining vector quantization with GANs and transformers for high-resolution image synthesis. Recent advancements in autoregressive image modeling include VAR, which employs transformer architectures to progressively predict higher-resolution token maps from lower-resolution inputs. Building upon this foundation, researchers have developed conditional VAR variants such as Controllable Autoregressive Modeling (CAR) [45] and ControlVAR [15]. These enhanced frameworks incorporate conditional mechanisms to facilitate controlled image generation through guided synthesis processes.

## 3 Method

### 3.1 Preliminary: Visual Autoregressive via Next-scale Prediction

Unlike traditional AR models, VAR operates on a multi-scale representation of an image, generating it from coarse to fine resolutions. An image is first encoded into a series of token maps $T = (t_1, t_2, \ldots, t_K)$ using a multi-scale VQVAE, where $t_k \in [V]^{h_k \times w_k}$ denotes the token map at scale $k$, with $h_k \times w_k$ increasing as $k$ grows from 1 (coarsest) to $K$ (finest). The auto-regressive likelihood is then expressed as

$$p(t_1, t_2, \ldots, t_K) = \prod_{k=1}^{K} p(t_k \mid t_1, t_2, \ldots, t_{k-1}), \quad (1)$$

where each $t_k$ is predicted based on all previous scales $t_1, \ldots, t_{k-1}$, and tokens within $t_k$ are generated in parallel.

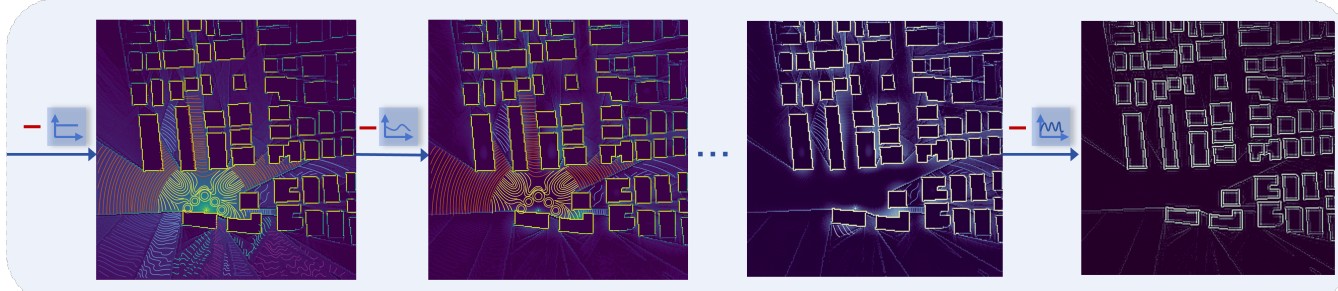

**Figure 3: Method motivation illustration. A case illustrates the Residual Laplacian Pyramid Decomposition's ability to cleanly separate low-frequency trends from high-frequency details in a radio map. The decomposition comprehensively captures multi-scale signal characteristics, from broad, low-frequency patterns to fine, high-frequency textures, demonstrating its effectiveness for hierarchical modeling of radio propagation.**

VAR utilizes a transformer architecture akin to GPT-2, as detailed in [22], employing a block-wise causal attention mask during inference to effectively manage multi-scale tokens. Unlike traditional methods that flatten token maps, VAR operates directly on 2D token structures, preserving spatial locality and maintaining the relationships between tokens. Additionally, by generating tokens in parallel within each scale, VAR reduces the number of autoregressive steps to $K$, where $K$ denotes the number of scales. This optimization lowers the computational complexity to $O(n^4)$ for an $n \times n$ image, significantly improving efficiency.

By generating token maps in a coarse-to-fine manner and parallelizing token generation within each scale, VAR provides an efficient alternative to raster-scan autoregressive decoding. Prior work also reports favorable scaling behavior and competitive zero-shot transfer in visual generation, which motivates adapting next-scale prediction to conditional radio map estimation.

### 3.2 Autoregressive Modeling for Accurate RME

Inspired by the visual autoregressive framework, we propose RadioAR, a next-scale prediction paradigm for radio map estimation. The overall framework of our model is illustrated in Fig. 4, offering a high-level depiction of its key components and their interactions. In this section, we present our problem formulation for conditional radio map estimation and introduce the architecture of our proposed model.

***Problem Formulation***. In this work, we address the task of conditional radio map estimation, where the goal is to predict a radio map $x \in \mathbb{R}^{H \times W}$ based on a two-channel input condition $C = (c_{\text{building}}, c_{\text{transmitter}})$. The input condition consists of:

- $c_{\text{building}} \in \{0, 1\}^{H \times W}$: a binary mask image representing the building shape, where white pixels indicate the presence of a building.
- $c_{\text{transmitter}} \in \{0, 1\}^{H \times W}$: a binary mask image representing the transmitter's position, featuring a single white pixel to precisely indicate the transmitter's location.

The generated radio map $R$ should accurately reflect the Received Signal Strength (RSS) across the area, accounting for the building structures and transmitter location specified in $C$.

---

**Algorithm 1** Residual Laplacian Pyramid Decomposition

---

**Input:** Latent image $z$, resolutions $(n_k)_{k=0}^{K}$
**Output:** Components $\{f_0, f_1, \ldots, f_K\}$
$g_0 = \text{downsample}_{1 \times 1}(z)$
$f_0 = \text{FEM}(g_0)$           ▷ DC component
$u_0 = \text{upsample}_{H \times W}(f_0)$
$\Delta z_0 = z - u_0$
**for** $k = 1$ to $K$ **do**
     $\mathbf{g}_k = \text{downsample}_{n_k \times n_k}(\Delta z_{k-1})$
     $f_k = \text{FEM}(g_k)$     ▷ Feature Enhancement Module
     $u_k = \text{upsample}_{H \times W}(f_k)$
     $\Delta z_k = \Delta z_{k-1} - u_k$
**end for**
**return** $\{f_0, f_1, \ldots, f_K\}$

---

***Multi-Scale Radio Map Tokenizer***. In autoregressive models for image generation, the tokenizer serves as a critical component, transforming complex image data into a sequence of tokens that can be processed sequentially by models like transformers. Unlike VAR's discrete tokenization via a multi-scale VQVAE, we employ continuous tokens to avoid quantization errors and instability, which are particularly problematic for radio maps with continuous signal patterns and sharp transitions. A VAE encoder encodes the radio map $x$ into the latent representation $z$, preserving spatial continuity and signal details critical for accurate radio map estimation, as confirmed in our ablation study. Furthermore, our VAE-based tokenizer inherently facilitates noise robustness. By mapping the radio map onto a compact latent representation $z$, the encoder learns to capture the underlying signal manifold. During this information bottleneck process, stochastic measurement noise and non-structural irregularities are effectively filtered out, ensuring that the autoregressive transformer operates on a denoised physical representation of the radio environment.

To capture multi-scale signal characteristics in radio maps, we draw inspiration from VAR's discrete multi-scale tokenizer and design the Residual Laplacian Pyramid Decomposition algorithm (algorithm 1) to align with our continuous token framework. We begin by simplifying VAR's multi-scale VQVAE Encoding process.

**Table 1: Performance comparison of various models for RME. Metrics include NMSE, RMSE, SSIM, PSNR, number of parameters (#Params) and inference time (Time).**

| Type | Model | #Params | NMSE ↓ | RMSE ↓ | SSIM ↑ | PSNR ↑ | Time |
|------|-------|---------|--------|--------|--------|--------|------|
| Conv. | RadioUnet [14] | 13.27M | 0.0135 | 0.0377 | 0.9133 | 28.52 | 0.004s |
| Conv. | EDSR-L32-D64 [35] | 2.40M | 0.0819 | 0.0807 | 0.7873 | 22.06 | 0.004s |
| Conv. | EDSR-L64-D64 | 4.77M | 0.0412 | 0.0571 | 0.8466 | 25.13 | 0.008s |
| Trans. | Radionet-L3-D128 [35] | 1.79M | 0.1427 | 0.1095 | 0.7246 | 19.40 | 0.002s |
| Trans. | Radionet-L3-D256 | 3.72M | 0.0301 | 0.0474 | 0.8439 | 26.80 | 0.003s |
| Trans. | Radionet-L6-D128 | 2.97M | 0.0334 | 0.0502 | 0.8396 | 26.29 | 0.004s |
| Diff. | RadioDiff-S50 [39] | 315.13M | 0.0195 | 0.0399 | 0.8923 | 28.16 | 4.572s |
| Diff. | RadioDiff-S1000 | 315.13M | 0.0165 | 0.0359 | 0.9101 | 29.13 | 75.692s |
| GAN | RME-GAN [49] | 13.27M | 0.0454 | 0.1076 | 0.8442 | 27.65 | 0.003s |
| Mamba | UVM-Net [50] | 1.01B | 0.0339 | 0.0492 | 0.8657 | 26.59 | 0.141s |
| AR | RadioAR-L16 | 290.11M | 0.0308 | 0.0459 | 0.8770 | 26.75 | 0.229s |
| AR | RadioAR-L24 | 778.18M | 0.0168 | 0.0382 | 0.9195 | 28.35 | 0.691s |
| AR | RadioAR-L30 | 1.43B | **0.0132** | **0.0331** | **0.9217** | **29.58** | 1.210s |

Specifically, applying a stripped-down version of algorithm 1—without the Feature Enhancement Module (FEM)—to a sample radio map reveals its ability to separate low-frequency trends from high-frequency details. For visualization, each residual $\Delta z_k$ is processed with a Laplacian sharpening filter to enhance texture features across different frequency scales, as demonstrated in a toy case (Fig. 3). This clearly demonstrates how $f_k$ captures progressively higher-frequency components, transitioning from broad, low-frequency patterns to intricate, high-frequency textures as the pyramid levels increase. This multi-scale decomposition hierarchically dissects the radio map into spatial frequency components, enabling detailed modeling of signal propagation.

To be specific, our complete decomposition processes the latent representation $z$ into components $F = (f_0, f_1, \ldots, f_K)$. It starts with the DC component $f_0$, obtained by downsampling $z$ to $1 \times 1$, representing the global signal baseline. Residuals $\Delta z_k$ are then recursively computed across increasing resolutions $n_k$ (where $n_0 = 1$, $n_k < n_{k+1}$), capturing finer details at higher scales. The Feature Enhancement Module (FEM) is a lightweight component designed to amplify subtle spatial variations. It consists of three residual convolutional layers with 64 channels and ReLU activations. By processing each downsampled residual $g_k$, the FEM refines the latent features across scales with minimal computational overhead, enhancing the overall decomposition precision.

Within our RadioAR framework, this multi-scale decomposition hierarchically models spatial frequency components of radio maps. Initial levels encapsulate coarse, low-pass structures, while higher-level residuals delineate fine, high-frequency variations, facilitating precise and comprehensive radio propagation modeling.

**Radio Autoregressive Transformer.** RadioAR leverages the conditional input $C$ to guide the estimation of the radio map $x$. $x$ represented as multi-scale token maps $F = (f_0, f_1, \ldots, f_K)$ by Multi-Scale Radio Map Tokenizer, with resolutions increasing from $k = 0$ (coarsest) to $k = K$ (finest). The conditional autoregressive likelihood is factorized as

$$p(F \mid C) = \prod_{k=0}^{K} p(f_k \mid f_{<k}, C), \qquad (2)$$

where $f_{<k} = (f_0, \ldots, f_{k-1})$ captures prior scales. To predict the token maps progressively, a Radio Autoregressive Transformer is employed, as shown in Fig. 4, starting from the smallest scale and refining details at higher resolutions. Additionally, the Conditional Integration Module consists of a Condition Encoder that maps the input condition $C$ to a conditional embedding $s$ and a DC Predictor that estimates the DC component (global signal baseline) from $C$. The training objective minimizes the negative log-likelihood:

$$\mathcal{L} = -\sum_{k=0}^{K} \log p(f_k \mid f_{<k}, C). \qquad (3)$$

The Radio Autoregressive Transformer is built upon the GPT-2 architecture, a transformer-based model widely recognized for its proficiency in autoregressive sequence modeling. After generating all token maps $F = (f_0, f_1, \ldots, f_K)$, the radio map $x$ is reconstructed using the decoder $D$ of the Multi-Scale Radio Map Tokenizer, as detailed in algorithm 2.

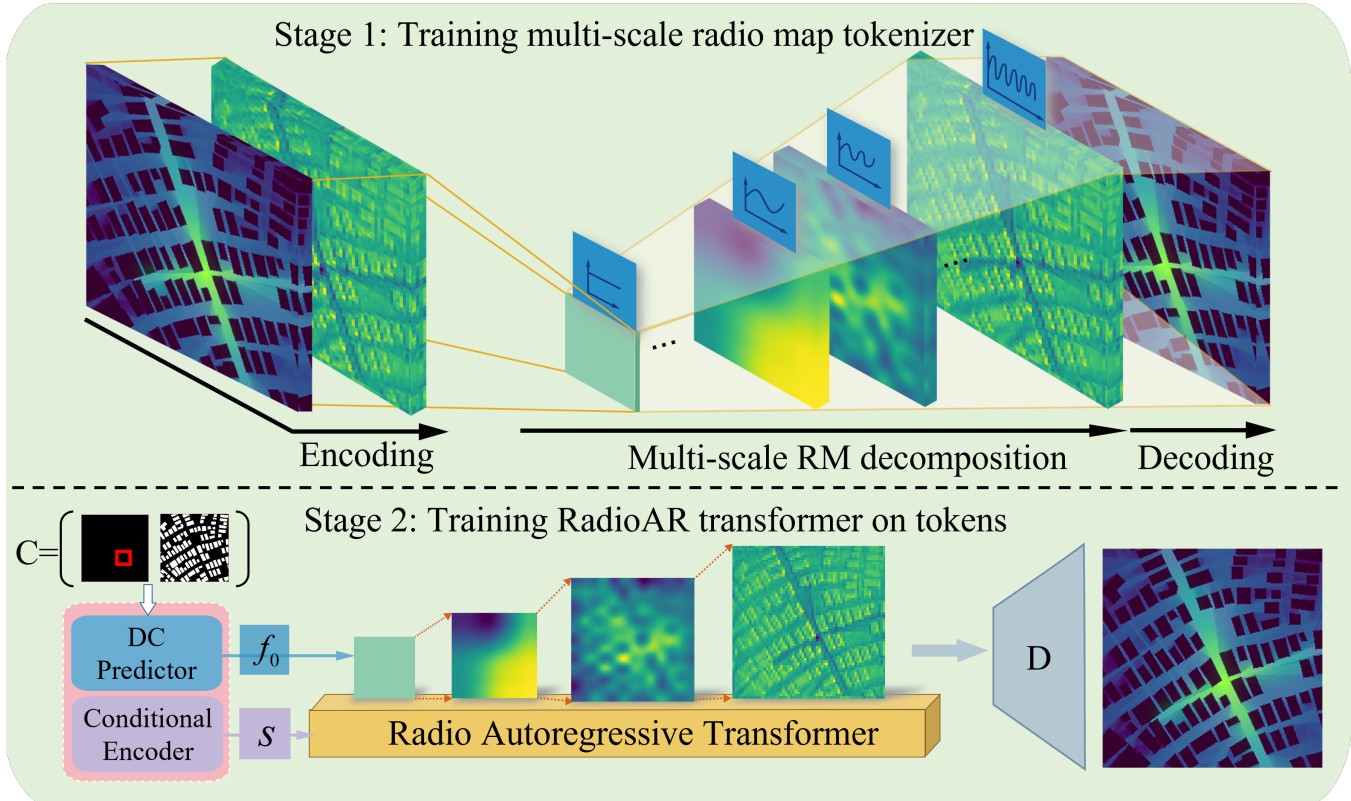

**Figure 4: Overview of our proposed RadioAR architecture for radio map estimation. Stage 1 illustrates the training of the multi-scale radio map tokenizer, encoding the input radio map into a latent representation and decomposing it into multi-scale token maps using a Residual Laplacian Pyramid. Stage 2 depicts the training of the RadioAR transformer, with the DC Predictor estimating the global signal baseline, the Conditional Encoder processing input conditions $C = (c_{\text{building}}, c_{\text{transmitter}})$ into an embedding, and the transformer progressively refining tokens into the final radio map.**

## 4 Experimental Results

### 4.1 Dataset

We use the RadioMapSeer dataset [46], which provides simulated path-loss radio maps for tasks including RSS radio map estimation and wireless localization. The maps are generated with Altair WinProp ray-tracing and cover 701 city regions extracted from OpenStreetMap (OSM) [19]; each region spans $256 \times 256$ meters at 1 m/pixel resolution.

We focus on the IRT4 subset, generated by the Intelligent Ray Tracing (IRT) model [42] with up to four interactions (reflections / diffractions). Transmitters are placed at 1.5 m height (one location per map). Maps are stored as PNG images where path-loss values are linearly mapped to grayscale intensities in [0, 255]. Unless stated otherwise, we follow the dataset configuration (transmit power 23 dBm, center frequency 5.9 GHz, bandwidth 10 MHz, noise figure 0 dB), and clip path loss below $-127$ dB.

### 4.2 Performance Evaluation

*Setup.* We evaluate RadioAR with depths 16/24/30 on the IRT4 subset of RadioMapSeer. Baselines include convolutional models (RadioUnet [14], EDSR-Lx-Dy [16]), transformers (Radionet-Lx-Dy [35]), diffusion (RadioDiff-S50 [39], RadioDiff-S1000), GAN (RME-GAN [49]), and Mamba (UVM-Net [50]). All experiments are conducted on an NVIDIA A40 GPU; inference time is measured with batch size 1. We train the tokenizer in two stages: (i) train a VAE to obtain a compact latent representation, and (ii) freeze the VAE and train the feature enhancement module (FEM) within the residual Laplacian pyramid to stabilize decomposition.

---

**Algorithm 2** Reconstruction of Radio Map from Components

**Input:** Components $\{f_0, f_1, \ldots, f_K\}$, original resolution $H \times W$
**Output:** Reconstructed radio map $\hat{x}$
Initialize $\hat{f} = 0$     ▷ Zero tensor of size $H \times W$
**for** $k = 0$ to $K$ **do**
 $u_k = \text{upsample}_{H \times W}(f_k)$
 $\hat{f} = \hat{f} + u_k$
**end for**
$\hat{x} = D(\hat{f})$   ▷ Decoder of multi-scale radio map tokenizer
**return** reconstructed radio map $\hat{x}$

---

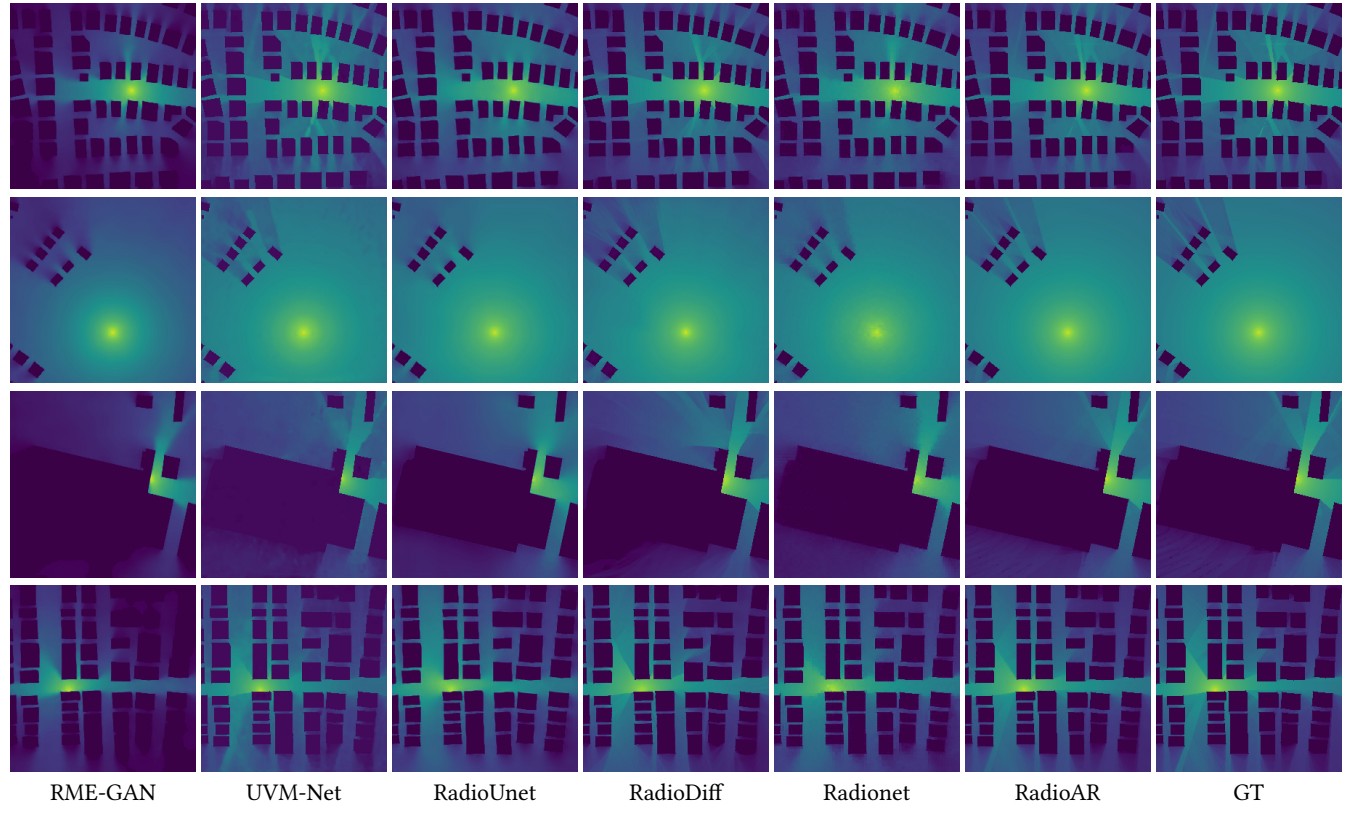

RME-GAN      UVM-Net      RadioUnet      RadioDiff      Radionet      RadioAR      GT

**Figure 5: Visual comparison of RME from RME-GAN [49], UVM-Net [50], RadioUnet [14], RadioDiff [39], Radionet (L3-D256) [35], and RadioAR (L30) against the GT across four test scenarios.**

*Estimation Accuracy Comparison.* The RadioAR models demonstrate superior estimation accuracy compared to state-of-the-art deep learning approaches for radio map estimation, as evidenced by comprehensive performance metrics. As presented in Tab. 1, RadioAR-L30 achieves performance comparable to the best convolutional models while surpassing them in key quality metrics, such as PSNR, indicating enhanced preservation of spatial details. In comparison to transformer-based models, RadioAR exhibits lower error rates, showcasing its ability to capture complex spatial patterns effectively. Diffusion-based models, while competitive in quality metrics, require significantly more computational resources, limiting their practicality. Even at shallower depths, RadioAR maintains robust performance, outperforming most baseline models across error and quality metrics. This consistent accuracy across varying model complexities underscores the effectiveness of the multi-scale autoregressive framework in balancing precision and generalization for radio map estimation tasks.

*Inference Efficiency Comparison.* The performance comparison illustrated in Tab. 1 highlights that RadioAR effectively balances inference speed and estimation precision, making it particularly suitable for real-time deployment in 6G networks. Its inference times, while higher than those of lightweight convolutional and transformer-based models, remain within practical limits for dynamic environments. In contrast, diffusion-based models exhibit significantly longer inference times due to their iterative sampling

**Table 2: Performance comparison of RadioAR variants in the ablation study for radio map estimation. Metrics include NMSE, RMSE, SSIM, and PSNR.**

| Description | Model | NMSE↓ | RMSE↓ | SSIM↑ | PSNR↑ |
|---|---|---|---|---|---|
| Discrete token | L16 | 0.0226 | 0.0503 | 0.8992 | 25.97 |
| +Scale up | L30 | 0.0217 | 0.0412 | 0.8931 | 27.68 |
| +Continuous token | L30 | **0.0132** | **0.0331** | **0.9217** | **29.58** |

processes, rendering them less viable for latency-sensitive applications. The progressive generation mechanism of RadioAR enables efficient computation, with inference times scaling predictably with model depth. This scalability ensures that RadioAR can deliver high-fidelity radio maps without excessive computational overhead. Compared to convolutional models, which prioritize speed but sacrifice accuracy, RadioAR provides an optimal trade-off. While our largest model (RadioAR-L30) defines the performance ceiling for RME, the framework is intrinsically scalable. For resource-constrained edge deployment, we consider research into knowledge distillation and pruning to transfer the high-fidelity spatial knowledge of the transformer decoder into compact student networks, a strategy that balances SOTA accuracy with real-time requirements in smart-city nodes.

***Visual Comparison.*** As illustrated in Fig. 5, our RadioAR (L30) model demonstrates superior visual fidelity in radio map estimation compared to baseline models, including RME-GAN, UVM-Net, RadioUnet, RadioDiff (S1000), and Radionet. The visual outputs of RadioAR closely resemble the Ground Truth (GT) across multiple test scenarios, exhibiting sharper boundaries and a more accurate representation of signal intensity variations. Convolutional models, such as RadioUnet, tend to produce smoother maps that may obscure fine-grained spatial details, while diffusion-based models, such as RadioDiff, achieve competitive visual quality but occasionally exhibit inconsistencies in signal continuity due to their iterative sampling nature. In contrast, the multi-scale autoregressive framework of RadioAR effectively captures both global structures and local details, resulting in visually coherent and precise radio maps.

## 4.3 Ablation Study

In this ablation study, we evaluate the impact of two critical components on the performance of RadioAR: model scaling and the choice between discrete and continuous tokens.

***Effect of Model Scaling.*** We assess the impact of model scaling by comparing RadioAR-L16 with discrete tokens to RadioAR-L30 with discrete tokens, as presented in Tab. 2. Scaling the model depth from 16 to 30 layers improves key performance metrics: NMSE decreases from 0.0226 to 0.0217, RMSE reduces from 0.0503 to 0.0412, and PSNR increases from 25.97 to 27.68. These enhancements reflect improved accuracy and signal quality attributable to the increased model capacity.

***Discrete vs. Continuous Tokens.*** We compare our RadioAR-L30 model using discrete tokens against its counterpart with continuous tokens to assess the impact of token representation on performance. Continuous tokens enable the model to better capture the inherently continuous nature of radio signal propagation, avoiding the quantization errors introduced by discrete representations. This results in enhanced precision across error metrics and improved quality in SSIM and PSNR. The adoption of continuous tokens allows for finer-grained modeling of spatial variations, which is critical for accurate radio map estimation in dynamic wireless environments.

## 5 Limitations and Future Work

While RadioAR achieves state-of-the-art performance on the RadioMapSeer (IRT4) benchmark, several limitations remain. First, our current evaluation relies on high-fidelity simulations; the transition to stochastic real-world measurements represents a primary research frontier. Second, the framework does not yet explicitly incorporate differentiable Maxwell-based physical priors within the loss function. Future work will focus on integrating these physical constraints and investigating model distillation to enhance scalability for diverse, resource-constrained 6G deployment scenarios.

## 6 Conclusion

We presented RadioAR, a multi-scale autoregressive approach to conditional radio map estimation. By bridging the gap between multi-scale prediction and continuous signal continuity, RadioAR ensures the integrity of the learned electromagnetic field, preserving critical multipath gradients that discrete methods often blur.

Our results demonstrate that RadioAR provides a favorable balance between accuracy and inference latency, supporting high-precision radio cartography in emerging wireless networks.

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
