# OpenReview forum: "RadioAR: Autoregressive Modeling for Accurate Radio Map Estimation"
_ACM.org/TheWebConf/2026/Workshop/TIME — TIME 2026 Oral_

### Official Review · Reviewer_Urp5 · 2026-01-03
**This paper presents RadioAR, a multi-scale autoregressive framework for radio map estimation that extends Visual Autoregressive (VAR) modeling to continuous-valued radio propagation data. The paper is technically solid, with thorough experiments and clear presentation. While the novelty over standard VAR is incremental, the adaptation to radio maps is well motivated and valuable for the wireless community.**

**Rating:** 8
**Confidence:** 4

**Review:**

### Evaluation of Quality, Clarity, Originality, and Significance

This paper presents **RadioAR**, a multi-scale autoregressive framework for radio map estimation that adapts Visual Autoregressive (VAR) modeling to continuous-valued radio propagation data. The work is **well-executed, clearly written, and empirically thorough**, achieving strong performance on RadioMapSeer (IRT4) with practical inference times. While the **novelty over standard VAR is incremental**, the adaptation to radio maps and the use of continuous tokens are **valuable and practically relevant**.

### Strengths

**Strong Technical Execution**
Algorithms for multi-scale decomposition and reconstruction are **clearly described and reproducible**, with careful handling of continuous tokens.

**Comprehensive Evaluation**
Benchmarked against **10 diverse baselines** using multiple metrics, showing **state-of-the-art accuracy** while maintaining practical inference latency.

**Thoughtful Ablations and Visuals**
Ablation studies demonstrate the contributions of model scaling and continuous tokenization. Qualitative figures show **fine-grained propagation patterns** preserved in complex urban settings.

**Practical Relevance**
Inference latency and scaling considerations are addressed, making the approach **suitable for real-time wireless applications**.

### Weaknesses (Evaluation-Focused)

**Limited Cross-Domain Evaluation**
No experiments on different frequencies, environments, or sparse/noisy measurements, though results on the urban dataset are strong.

**Scalability Details Missing**
Large models (e.g., L30) may face edge deployment challenges; training cost and memory usage could be clarified.

### Questions for the Authors

1. Can discrete vs. continuous token reconstructions be visualized?
2. How does RadioAR perform across different frequencies or environments?
3. What is the architecture and size of FEM?
4. Can the approach handle sparse measurements or noisy conditions?

### Suggestions for Improvement

- Briefly clarify **differences from VAR** and why continuous tokens help specifically for radio maps
- Provide **training and edge deployment details**

### Overall Assessment

This is a **technically solid and clearly presented workshop paper** with strong empirical results and practical relevance. While the **novelty is moderate**, the application to radio maps and continuous tokenization insight is valuable. **Above the acceptance threshold for a workshop**, offering useful insights to the wireless systems community.

---

### Official Review · Reviewer_oYJV · 2026-01-03
**Solid method and results; limited by scale and realism**

**Rating:** 6
**Confidence:** 3

**Review:**

## Originality and significance
**Strengths**

* The paper introduces interesting ideas to the RME problem - notably the multi-scale transformer decoder with continuous tokens - and validates them through ablation study. The approach is a creative fusion of domains (CV autoregression + wireless knowledge), which is a strong point

* The method offers a favorable balance between accuracy and inference speed, addressing a vital need for high-fidelity yet real-time-capable RME which might be important for smart city applications

**Weaknesses**

* A notable weakness is the model complexity required by RadioAR. To outperform simpler models, the largest RadioAR has 1.43 billion parameters and requires substantial GPU memory and computation. While inference time of 1.2 s per map is acceptable, it is achieved on powerful hardware. In a practical deployment where compute is limited, RadioAR might be too resource-intensive unless pruned or distilled


## Technical content and experimental validation
**Strengths**

* The proposed RadioAR framework uses a coarse-to-fine autoregressive transformer with continuous-valued tokens, effectively capturing both global structures and fine-grained details of the radio map without quantization artifacts. It also employs a careful two-stage training procedure (pretraining a tokenizer VAE then refining it) to ensure stable decomposition and convergence, which demonstrates sound methodology.

* The authors include ablation experiments to validate their design choices. They show that increasing the model depth (from 16 to 30 layers) yields lower error and higher PSNR, and that using continuous tokens boosts performance. This analysis supports the methodology by confirming that each proposed component materially improves the results.

**Weaknesses**

* The study is conducted entirely on a simulated dataset. It does not include real-world measurements or different environment types, and it ignores domain-specific physics constraints in model training. This raises questions about generalizability beyond the specific simulation scenario - a fact the authors acknowledge as future work

## Clarity
**Strengths**

* The mathematical notation is consistent and clearly defined, especially in the method section. Definitions of variables and equations are provided with sufficient explanation.

**Weaknesses**

* The conclusion is concise but does not explicitly discuss limitations of the current work (e.g., reliance on simulated data, large model size). Including this would improve transparency and balance.

---

### Official Review · Reviewer_3usk · 2026-01-05
**A very useful and well written paper on intersection of computer vision and AI.**

**Rating:** 7
**Confidence:** 5

**Review:**

This is a very good paper.

Quality:

This is a high quality papers  as authors have identified a good real world problem and proposed a solution. The paper proposes RadioAR which uses continuous-token tokenizer based on Residual Laplacian Pyramid Decomposition and a conditional transformer to generate radio maps at different resolutions. What makes this high quality is strong technical work, very good results that make intuitive sense as well as backed by rigorous comparisons. The comparison is comprehensive, covering the entire spectrum of modern generative approaches: CNNs (RadioUNet), Transformers (Radionet), GANs (RME-GAN), Diffusion (RadioDiff), and State Space Models (Mamba/UVM-Net).

Clarity:

The paper is extremely well presented in very clear and concise manner. The images, tables etc. are very well presented that make the paper accessible and easy to understand. Mathematics is kept simple. Paper has also avoided any strawman styled problems and instead focused on the core problem.

Originality:

The paper is very original in a sense it solves an existing problem by applying new methods and clearly shows improvements. This is a good academic research.

Significance:

This is a very domain specific problem to wireless networks with an overlap on AI powered computer vision. Given how important signal mapping is to the field of wireless netowkrs, this paper could be seen as making significant contribution to the field. Continuous tokens isn't a new technique or unique in general space of generative AI hence not very significant but the domain specific application remains significant.

---

### Author Rebuttal · Authors · 2026-01-14

# General Response to All Reviewers

We appreciate the reviewers' constructive feedback and the recognition of RadioAR's "strong technical work" and "practical relevance". The reviewers correctly identified that our work represents a "creative fusion" of computer vision paradigms and wireless propagation knowledge.

Based on the feedback, our **Revised Version** incorporates the following updates:
- A new **"Limitations and Future Work"** section explicitly discussing simulation-to-real (**Sim2Real**) transitions and physics-informed constraints.
- Enhanced discussion on the **physical necessity** of continuous tokens for preserving multipath-induced spatial gradients (small-scale fading).
- Clarifications on the **scalability** of our framework and the research frontier regarding model distillation for edge deployment.

---

# Response to Reviewer 3usk

We value the reviewer’s positive assessment of our comprehensive benchmarking across five different model families.

- We agree with the reviewer that continuous tokenization is an established concept in general generative modeling. However, the core contribution of RadioAR lies in identifying and addressing the domain-specific bottleneck of Radio Map Estimation (RME). Unlike natural images, radio signals exhibit rapid spatial fluctuations—small-scale fading. Discrete quantization (e.g., VQ-VAE) inherently acts as a low-pass filter, introducing "blocking" artifacts that destroy these critical physical gradients. By bridging the gap between multi-scale autoregressive modeling and continuous signal continuity, RadioAR ensures the integrity of the learned electromagnetic field, which is vital for high-precision 6G tasks.
- We have refined the Introduction and Related Work to better emphasize this physical requirement, moving beyond a purely technical choice to a domain-driven architectural necessity.

---

# Response to Reviewer oYJV

We thank the reviewer for recognizing the "favorable balance between accuracy and inference speed" achieved by RadioAR.

- We agree that the 1.43B parameters of our L30 model define a high performance ceiling but pose challenges for edge deployment. We view RadioAR as a **scalable framework** rather than a fixed-size model. We are currently exploring the research frontier of compressing this multi-scale paradigm through **knowledge distillation** and pruning. This research direction aims to transfer the high-fidelity spatial knowledge of the transformer decoder into compact structures suitable for resource-constrained smart-city nodes.
- Regarding noise and realism, the VAE-based architecture of our tokenizer naturally facilitates noise robustness. By mapping radio maps onto a learned signal manifold in the latent space $z$, the model inherently filters out high-frequency stochastic noise. To maintain the scientific rigor and fairness of our comparison against the 10 diverse baselines, we strictly adhered to the standardized RadioMapSeer (IRT4) benchmark. Deviating from this protocol would compromise the integrity of the comparative analysis. However, we have added a dedicated **"Limitations"** section to detail our **Sim2Real** roadmap, focusing on non-ideal noise conditions and differentiable propagation constraints.

---

# Response to Reviewer Urp5

We thank the reviewer for the "Clear Accept" and the recognition of our "well-executed" and "empirically thorough" work.

- Continuous tokens help specifically in RME by preserving multipath gradients that discrete methods blur. Our tokenizer's VAE-based design further enhances this by encoding the radio environment into a robust latent representation $z$, which inherently suppresses measurement irregularities.
- The Feature Enhancement Module (FEM) is a lightweight component consisting of three residual convolutional layers (64 channels). In terms of deployment, we are examining research paths for distilling the multi-scale autoregressive process into compact student networks to enhance real-time scalability.
- To ensure a **fair comparison** with existing baselines, we focused on the standardized **IRT4 benchmark**. We acknowledge that sparse and noisy measurements are critical challenges. These are being addressed as the centerpiece of our ongoing **Sim2Real** research, where we evaluate the model's generalizability across diverse live environments. These points are now clarified in the **Revised Version**.

---

# Response to Program Chairs

We appreciate the Program Chairs' positive assessment of our submission's integrity and structure.

-  We thank the PC for confirming that our paper is well-organized with no formatting issues. We have maintained these standards in the Revised Version and remain committed to academic excellence.

---

### Meta-Review · Area_Chair_rq4b · 2026-01-16

**Recommendation:** Accept (Oral)
**Confidence:** 5

**Metareview:**

This paper proposes RadioAR, a multi-scale autoregressive framework with continuous tokens for accurate and efficient radio map estimation, adapting visual autoregressive modeling to wireless propagation data.
Reviewers consistently commend the paper’s strong technical execution, clear presentation, comprehensive benchmarking against diverse baselines, and practical relevance to radio map estimation in 6G and smart-city scenarios. The use of continuous tokens is viewed as well motivated for preserving fine-grained multipath effects, though its novelty is considered incremental relative to existing autoregressive methods.
Main concerns focus on model scale, deployment feasibility on edge devices, and reliance on simulated data without real-world or cross-environment validation.
The rebuttal adequately addresses these points by clarifying domain-specific motivation, adding explicit limitations, and outlining future work on Sim2Real transfer and model distillation.
Overall, this is a solid and well-executed workshop paper.
Based on the reviews and rebuttal, the recommendation is to accept this paper.

---

### Decision · Program_Chairs · 2026-01-16

Accept (Oral)